# Synergistic effect between plant extracts and fluoride to protect against enamel erosion: An in vitro study

**Thiago Saads Carvalho** [1]* , **Khoa Pham** [1], **Daniela Rios** [2], **Samira Niemeyer** [1],
**Tommy Baumann** [1]

**1** Department of Restorative, Preventive and Pediatric Dentistry, School of Dental Medicine, University of Bern, Bern, Switzerland, **2** Department of Pediatric Dentistry, Orthodontics and Public Health, Bauru School of Dentistry, University of São Paulo, Bauru, São Paulo, Brazil

☉ These authors contributed equally to this work.
* thiago.saads@zmk.unibe.ch

**Data Availability Statement:** All relevant data are within the manuscript and its Supporting Information files.

## Abstract

Polyphenol-rich solutions, such as plant extracts and teas, can modify the salivary pellicle and improve the protection against dental erosion. In this study, we further explored how these polyphenol-rich plant extracts solutions behave in the presence of fluoride. We distributed enamel specimens into 9 groups (n = 15): Control_No_F⁻ (Deionized water); Control_F⁻ (500 ppm F-), Grape_Seed_No_F⁻ (Grape seed extract), Grape_Seed_F⁻ (Grape seed extract + 500 ppm F⁻), Grapefruit_Seed_No_F⁻ (Grapefruit seed extract), Grapefruit_Seed_F⁻ (Grapefruit seed extract + 500 ppm F⁻), Blueberry_No_F⁻ (Blueberry extract), Blueberry_F⁻ (Blueberry extract + 500 ppm F⁻), and $Sn^{2+}$/F⁻_Rinse (commercial solution containing 800 ppm $Sn^{2+}$ and 500 ppm F⁻). The specimens were submitted to 5 cycles (1 cycle per day), and each cycle consisted of: salivary pellicle formation (human saliva, 30 min, 37˚C), modification of the pellicle (2 min, 25˚C), pellicle formation (60 min, 37˚C), and an erosive challenge (1 min, citric acid). Between cycles, the specimens were kept in a humid chamber. Relative surface hardness (rSH), relative surface reflection intensity (rSRI) and calcium released to the acid were analysed, using general linear models, and Kruskal-Wallis with post-hoc Dunn's tests. We observed that the presence of fluoride in synergy with the extract solutions provided better protection than the groups containing extract or fluoride only. For rSH, we observed a significant main effect of extracts ($F_{(4,117)} = 9.20$; $p<0.001$) and fluoride ($F_{(1,117)} = 511.55$; $p<0.001$), with a significant interaction ($F_{(3,117)} = 6.71$; $p<0.001$). Grape_Seed_F⁻ showed the best protection, better than fluoride, and $Sn^{2+}$/F⁻_Rinse. Calcium results also showed greater protection for the groups containing fluoride, whereas for rSRI, despite a significant interaction between extract and fluoride ($F_{(3,117)} = 226.05$; $p<0.001$), the differences between the groups were not as clearly observed. We conclude that polyphenols from plant extracts, when combined with fluoride, improve the protective effect of salivary pellicles against enamel erosion.

**Funding:** The author(s) received no specific funding for this work.

**Competing interests:** The authors have declared that no competing interests exist.

## Introduction

The benefits of fluoride in dental caries are more than well-established in Dentistry, but its role in dental erosion is still very much debated. Under the right conditions, remineralization occurs when fluoride adsorbs onto a partially demineralized crystalline surface forming a fluor(hydroxy) apatite lattice. This usually occurs in subsurface of initial caries lesions, but during the remineralization process, entirely new crystals are not commonly formed [1]. Since dental erosion is the demineralization of the tooth surface, and the crystals are lost layer by layer, new crystals are not formed, and any remineralization is limited only to the very thin demineralized surface.

Rather than remineralization, the protection of fluoride against erosion is believed to be related to the formation of $CaF_2$-like particles on the tooth surface, which serve as a physical barrier against the acid challenges [2]. This depends on the fluoride concentration and pH of the product. However, even under optimized conditions, the formation of $CaF_2$-like particles does not completely cover the tooth surface, and the protection from monovalent fluoride products in erosion prevention might be of limited outcome [3,4]. On the other hand, products containing polyvalent metal ions, such as stannous, are able to form an acid resistant layer on the tooth surface or incorporate into the demineralized layer of the tooth [5,6]. This provides better protective effects than the monovalent form of fluoride products [7,8].

The salivary pellicle is also a natural protection against erosive demineralization. This layer, consisting of proteins, peptides, lipids and other macromolecules mainly from saliva, serves as a barrier against acids. The pellicle is made up of a dense basal layer adhered to the tooth structure via electrostatic and van der Waals interactions [9,10], and a more loose globular layer, linked together by protein-protein interactions resulting in globular and granular structures [11]. The basal layer is more resistant to dissolution by acids [12] and it accounts for the erosion-protective effect [13].

Both fluoride and stannous ions modulate and modify the salivary pellicle [14–16], enhancing its protective effect. Similarly, polyphenol-rich plant extracts modify the pellicle. The polyphenols cross-link the salivary proteins, increasing the protein content of the pellicle, leading to a thicker and more electron-dense basal layer that better withstands acid challenges [17,18]. Combining the pellicle modification properties of the polyphenols with the effect of fluoride on the tooth surface and the pellicle might, therefore, increase the protective effect against dental erosion. So, this study aimed at verifying how solutions containing both polyphenol-rich plant extracts and fluoride protect against enamel erosion in the presence of the salivary pellicle.

## Material and methods

### Experimental design

The present experiment was made with 135 enamel specimens that were distributed into 9 groups, according to the experimental solution used. We initially measured the baseline surface reflection intensity and surface hardness of all enamel specimens. The specimens were then individually submitted to 5 erosion cycles (one cycle per day), consisting of salivary pellicle formation (200 µl, 30 min, 37˚C, no movement), modification of the pellicle with the experimental solutions (10 ml, 2 min, 25˚C, 70 rpm, travel path 50 mm) and again pellicle formation (200 µl, 60 min, 37˚C, no movement). Between these steps, the specimens were washed with deionized water (10 s), dried with air (5 s). Lastly, after pellicle formation and modification, the specimens were submitted to an erosive challenge (10 ml of 1% citric acid, 1 min, pH 3.6, 25˚C, 70 u/min, travel path 50 mm). The specimens were washed again and stored in a humid chamber until the next cycle, beginning with salivary pellicle formation

once again. Between the cycles, the specimens were kept in a humid chamber. After each erosive challenge, the citric acid was stored for later analysis of calcium concentration, and after all cycles, surface hardness and surface reflection intensity were measured again.

## Preparation of the enamel specimens

From a pool of extracted human teeth, 135 enamel specimens were prepared. Because the teeth were kept in a pooled biobank, the local ethic committee (KEK) categorizes them as irreversibly anonymized, and no previous ethical approval was necessary. In accordance with the guidelines and regulations of the KEK, the volunteers were informed about the use of their teeth in research and their oral consent was obtained. The teeth were cut using an Isomet® low speed saw (Isomet, Buehler Ltd., Düsseldorf, Germany) and buccal and oral halves were embedded in acrylic resin (Paladur ®; Heraeus Kulzer GmbH, Hanau, Germany). The specimens were then ground using a Knuth Rotor machine (Labpol 21, Struers, Copenhagen, Denmark) with silicon carbide paper discs of grain size 18.3 μm, 8 μm, 5 μm for 60 seconds each. Between these steps, the specimens were rinsed and sonicated for 1 minute in water. Further polishing with 3 μm diamond abrasive was carried out for 60 seconds (LaboPol-6, DP-Mol Polishing, DP-Stick HQ, Struers, Copenhagen, Denmark). The flat and polished enamel specimens were stored in a mineral solution (1.5 mml/l $CaCl_2$, 1.0 mmol/l $KH_2PO_4$ 50 mmol/l NaCl, pH = 7.0) until the time of the experiment.

Immediately prior to the start of the experiment, the specimens were submitted to a final polishing with a 1 μm diamond abrasive paste under constant cooling (LaboPol-6, DP-Mol Polishing, DP-Stick HQ, Struers, Copenhagen, Denmark) and sonicated for 1 minute in water. Each enamel specimen was labelled with a number and randomly distributed into 2 main groups: presence or absence of fluoride, and each group was further divided into 4 subgroups, according to the polyphenol-rich plant extract. An additional subgroup corresponded to a positive control group (commercial solution containing stannous). Therefore, a total of 9 subgroups, each containing n = 15 specimens, were included in the study.

## Experimental solutions

The tested solutions were: Control_No_$F^-$ (Deionized water); Control_$F^-$ (500 ppm $F^-$), Grape_Seed_No_$F^-$ (Grape seed extract), Grape_Seed_$F^-$ (Grape seed extract + 500 ppm $F^-$), Grapefruit_Seed_No_$F^-$ (Grapefruit seed extract), Grapefruit_Seed_$F^-$ (Grapefruit seed extract + 500 ppm $F^-$), Blueberry_No_$F^-$ (Blueberry extract), Blueberry_$F^-$ (Blueberry extract + 500 ppm $F^-$), and $Sn^{2+}/F^-$_Rinse (commercial solution containing 800 ppm $Sn^{2+}$ and 500 ppm $F^-$). All solutions containing fluoride had a concentration of 500 ppm $F^-$, either from sodium fluoride (in all experimental solutions) or from a mixture of sodium fluoride and amine fluoride (as in the commercial solution). A previous study from our group used solutions with a polyphenol concentration of 1 g/l [17], where pellicle was modified for 30 min. In the present study, we reduced the modification time, and increased the polyphenol concentration to 2 g/l. The solutions were prepared (according to the manufacturers' declaration of polyphenol content), by dissolving the contents (powder) of the extract capsules in deionized water, mixing for 30 min at room temperature and then filtering. All solutions had their pH set to 5.8 by adding 1M KOH or 1M HCl, accordingly, except the commercial solution, which maintained its native pH of 4.5.

## Saliva collection

Whole mouth stimulated human saliva was collected from 30 healthy volunteers aged 20–30 years and from both genders, who were instructed not to eat or drink (except water) for 2 h

before the saliva collection, which was performed in the morning. Participants chewed on a piece of paraffin (Paraffin Pellets, Ivoclar Vivadent, Schaan, Liechtenstein) for 10 min, and the saliva was collected in chilled flasks. The saliva from all volunteers was then pooled, centrifuged (20 min; 4˚C; 3,000 g) and the supernatant was divided in small aliquots and stored at -80˚C until the time of experiment. Because the saliva was pooled, the local ethic committee (KEK) categorizes it as irreversibly anonymized, and no previous ethical approval was necessary. In accordance with the guidelines and regulations of the KEK, the volunteers were informed about the use of their saliva in research and their verbal consent was obtained.

## Surface hardness measurements

Surface hardness (SH) measurements were performed using a Vickers diamond under a load of 50 mN (5.1 grams-force) and a dwell of 15 s (Fischerscope HM 2000 XYp; Helmut Fischer, Hünenberg, Switzerland), at baseline and after all cycles. For each measurement, six indentations were made on the enamel surface at distance intervals of 25 μm. The average value from the 6 indentations was considered as the hardness value for the specimen. The relative hardness (rSH) was calculated using the formula $rSH = (SHfinal / SH_{baselline}) \times 100$, where $SH_{baselline}$ is the initial hardness measured at baseline (before any erosive challenge) and SHfinal is the value after the final experimental cycle.

## Surface reflection intensity measurements

Surface reflection intensity (SRI) measurements were performed with the Tabletop reflectometer at baseline (SRIbaseline) and at the end of the experiment (SRIfinal). Immediately prior to final measurement, the specimens were placed in sodium hypochlorite and incubated for 2 min to remove the salivary pellicle, so that it would not interfere with the readings. To perform the SRI measurements, the specimens were individually placed on a platform under a laser beam (635 nm; oeMarket, Cherrybrook, Australia), and the platform was then moved to adjust the laser beam and reach the highest reflection point. The reflected light was captured and measured with a photodiode (FDS100; Thorlabs, Dachau, Germany). The point of highest reflection intensity was then registered. Since the enamel specimens were highly polished at the start of the experiment, the highest SRI values correspond to non-eroded polished enamel surface, and as the experiment progressed, the value decreased. Thus, lower values signify greater erosive demineralization. For analyses, we calculated the relative surface reflection intensity (rSRI) using the formula: $rSRI = (SRIfinal / SRIbaseline) \times 100$.

## Amount of calcium released in citric acid

The amount of calcium released to the citric acid (CaR) was determined at the end of the experiment using an atomic absorption spectrometer (AAnalyst 400, Perkin Elmer Analytical Instruments, Waltham, MA, USA). The amount of citric acid used for each specimen in the 5 cycles (total of 50 ml per specimen) was pooled and used for the analysis. To eliminate the interference of other ions, lanthanum nitrate (0.5%) was added to the citric acid. The calcium concentrations were then normalized to the surface areas of the enamel specimens. The surface area of each specimen was measured with a light microscope (Leica, M420) connected to a camera (Leica, DFC495). Under 16 x magnification and using the software program IM500, the contour of the exposed enamel area was traced, and the enamel area calculate. The cumulative amount of calcium released during the erosive challenges was considered for the statistical analyses, which represents the total amount of calcium released by each specimen after all cycles and is expressed in nmol of $Ca^{2+}$ / $mm^2$ of enamel.

### Statistical analysis

Initially, the data were checked for normality with the Shapiro-Wilk test. Because some groups did not follow a normal distribution, these groups were checked for outliers. Also, to maintain the balance of equal number of specimens per group, random numbers were generated to select specimens to be removed from the other groups. In total, 1 specimen from each was removed, leaving the groups with n = 14 specimens each (overall 126 specimens) for the analysis. After removal of specimens, all groups were normally distributed for all variables, except one group for calcium release. So, for SH and SRI, parametric tests were performed, and for calcium release, a non-parametric analysis was used.

For SH and SRI, the effect of extracts and fluoride were initially verified with general linear models having extracts and presence of fluoride as factors, and a post-hoc Tuckey's test (Levene-test for homogeneity of variances for SH and SRI were p = 0.194 and p = 0.417, respectively).

For calcium release, a non-parametric equivalent with Kruskal-Wallis and post-hoc Dunn's tests were performed.

All analyses considered a significance level of 0.05.

## Results

In rSH (Fig 1), we observed a significant main effect for extracts ($F_{(4,117)}$ = 9.20; p<0.001), and likewise for fluoride ($F_{(1,117)}$ = 511.55; p<0.001). There was also a significant interaction

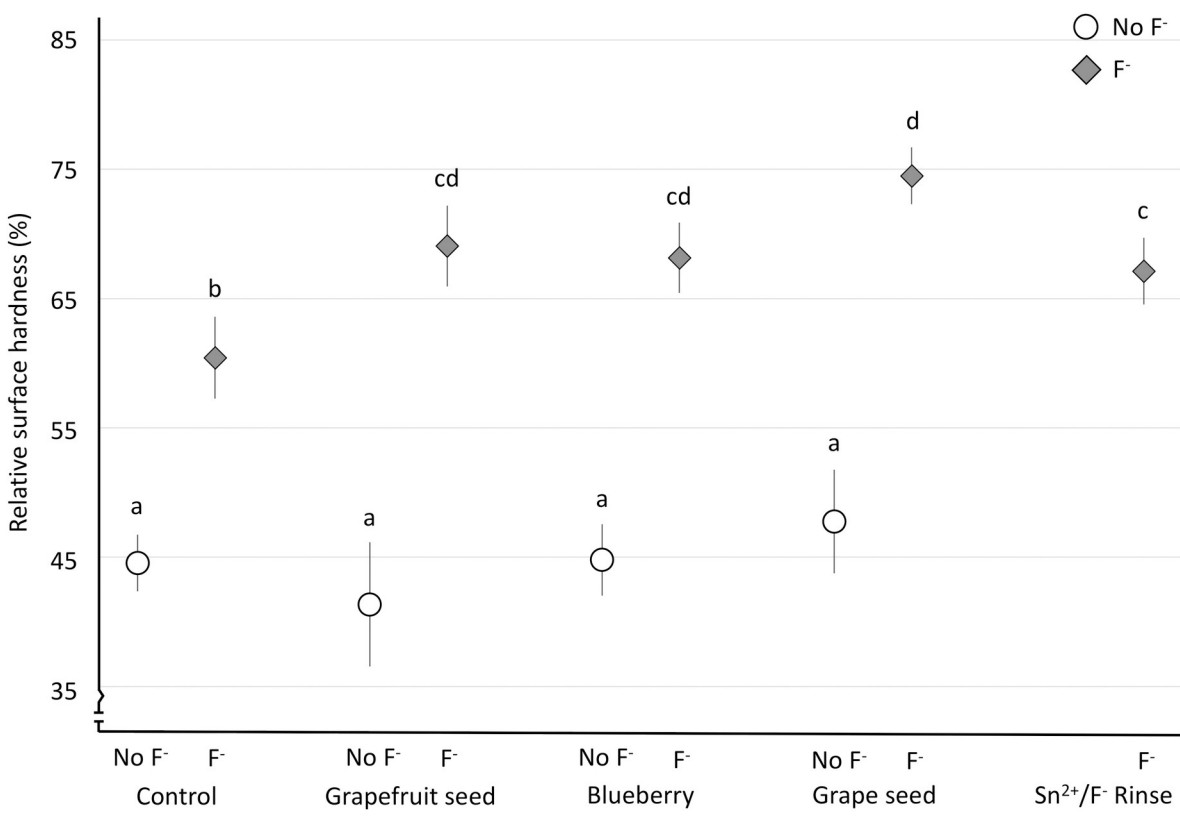

**Fig 1. Mean relative surface microhardness (rSH) and standard deviation (error bars), according to the solutions.** Groups with different letters are significantly different from each other. Different shaded symbols mean presence (gray diamonds–F⁻) or absence (white circles–No_F⁻) of fluoride.

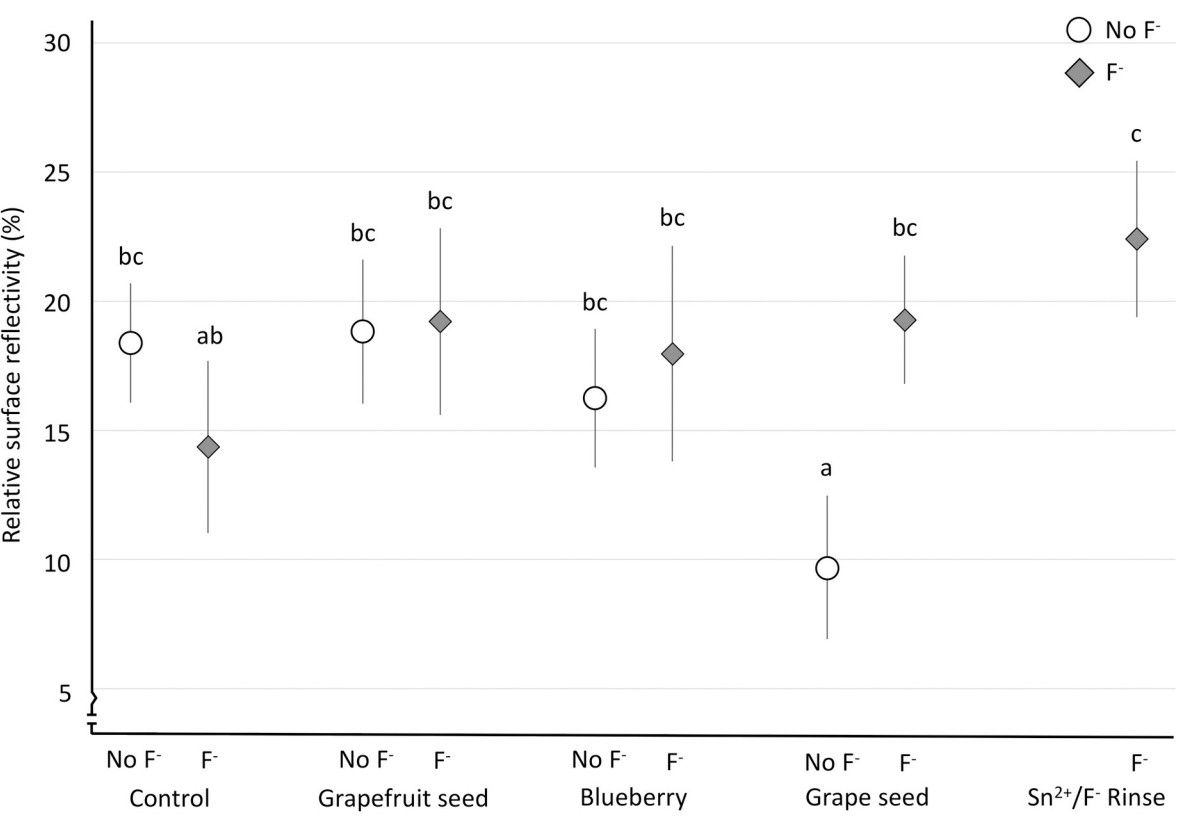

**Fig 2. Mean relative surface reflection intensity (rSRI) and standard deviation (error bars), according to the solutions.** Groups with different letters are significantly different from each other. Different shaded symbols mean presence (gray diamonds–F⁻) or absence (white circles–No_F⁻) of fluoride.

between extract and fluoride (F(3,117) = 6.71; p<0.001). In Fig 1, the clear distinction between the groups without and with fluoride is seen, where the latter provided more protection than the former. Moreover, the presence of both fluoride and extracts provided a significantly better protection than the group containing only fluoride (Control_F⁻). The group Grape_Seed_F⁻ showed the best protective effect, significantly better than both control groups (Control_No_F⁻ and Control_F⁻), as well as the Sn²⁺/F⁻_Rinse, but not different to the other extract groups with fluoride (Grapefruit_Seed_F⁻ or Blueberry_F⁻).

For rSRI (Fig 2), we observed a significant main effect of extracts (F(4,117) = 136.16; p = 0.001), but not significantly for the presence of fluoride (F(1,117) = 103.75; p = 0.058). In contrast, there was a significant interaction between extract and fluoride (F(3,117) = 226.05; p<0.001), though the differences between the groups were not as clearly observed (Fig 2).

Fig 3 differs from the others (boxplots and medians), because the calcium release data were not normally distributed, and a non-parametric analysis was carried out. Similar to the other parameters, the calcium release data show greater protection for the groups containing fluoride, where they generally released less calcium than those without fluoride (Fig 3). The best protection (less calcium release) was observed for the Sn²⁺/F⁻_Rinse group, similar to Grape_-Seed_F⁻, Grapefruit_Seed_F⁻ and Control_F⁻. In general, the groups containing extract and fluoride performed better than those containing just the extract (p<0.05), except for Blueberry, where there was no difference between Blueberry_No_F⁻ and Blueberry_F⁻.

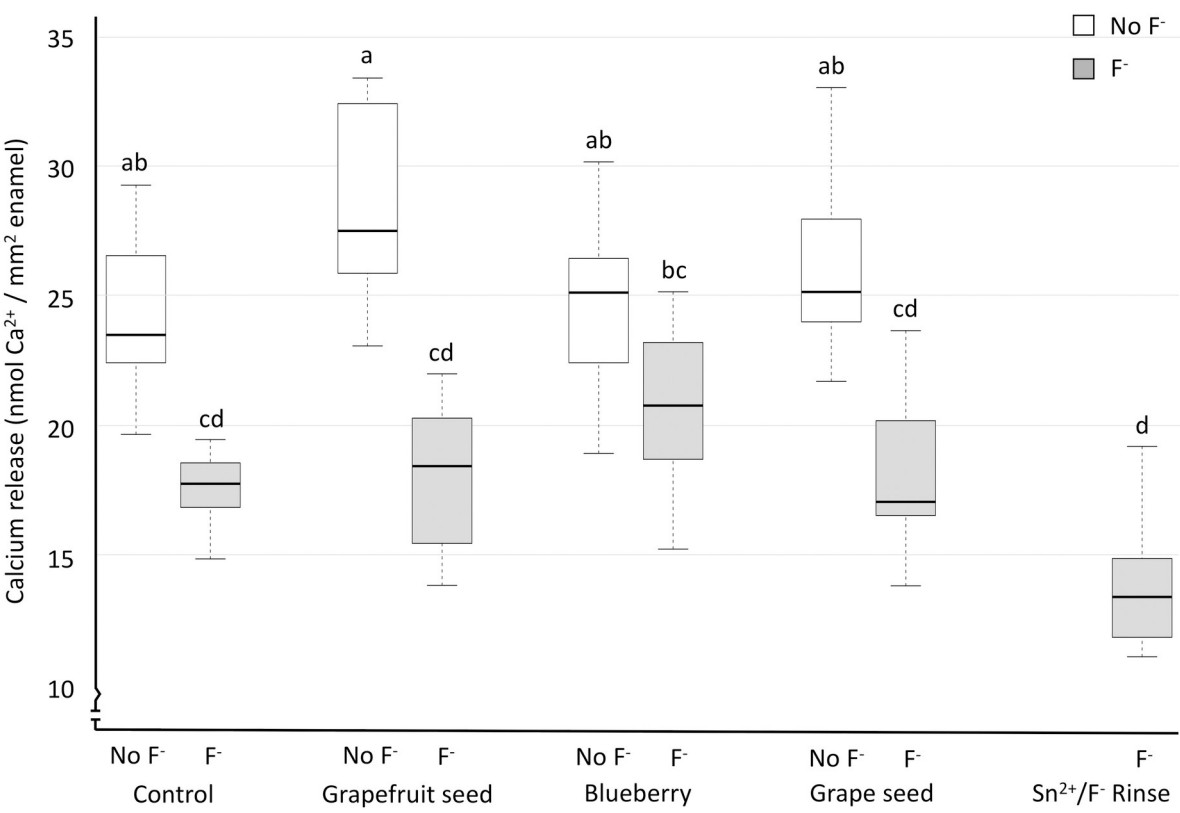

**Fig 3. Boxplots for total calcium released to the citric acid, according to the solutions.** The boxplots better portray non-parametric data, where the boxes represent interquartile range, dark bars represent median, and the whiskers represent highest and lowest readings. Groups with different letters are significantly different from each other. The gray shaded boxes represent groups with the presence (F⁻) of fluoride, and the white boxes represent absence (No_F⁻) of fluoride.

## Discussion

The present study tested different polyphenol-rich plant extracts with and without fluoride, and we observed significant differences between the groups. The differences were most clear in the rSH results, where the presence of fluoride in the solutions significantly protected against enamel erosion, but the results were not so clear-cut with the rSRI or calcium methods. This is probably because we included the salivary pellicle in our experimental design, which exerts some influence on the SRI and calcium methods.

Regarding the rSRI method, the pellicle is able to cover the microscopic grooves on the dental surface created after an erosive challenge [19], and even when NaOCl is used to remove the pellicle, remnants of the basal layer can still persist. These remnants would be removed if combining the NaOCl with ultrasound [20], but ultrasonication would also remove parts of the softened eroded enamel [21], so this was not done in this experiment. Therefore, the remnants of the basal layer left on our specimens increased the rSRI readings; and this especially befalls the groups treated with polyphenols, where the latter persists adsorbed onto the pellicle even after exposure to surfactants [22,23].

Regarding the calcium method, the pellicle also contains calcium, which is also released during the erosive challenge [24,25]. The calcium results in this case then refer to calcium released from both the dental surface as well as from the pellicle. The groups where the pellicle was modified with polyphenols did not differ from their control counterpart (with or without fluoride). It is speculated that the pellicle modified with polyphenols is denser and more

resistant to acid attacks, thus probably releasing less calcium. Since the calcium results are influenced by the presence of pellicle and its different modifications, the rSH method is probably the parameter that best represents the differences between the groups.

The rSH results, showed a clear positive effect from the presence of fluoride in the experimental solutions. In dental erosion, the main protective effect of fluoride is related to the $CaF_2$-like particles serving as a physical barrier that protects the underlaying tooth surface against the acid challenges [2]. These particles, however, are not acid-resistant, limiting the protective effect of fluoride. So, the debate on the effectiveness of fluoride in erosion is still in progress [3,26,27]. A more comprehensive protection would only be obtained if the $CaF_2$-like layer is dense enough to thoroughly cover the tooth surface, but even under ideal conditions with fluoride concentrations similar to that of toothpastes, the formation of the $CaF_2$-like layer covers less than 40% of enamel surface [4]. Interestingly, in the present study, the 500 ppm fluoride solution presented significantly better protection in comparison to no treatment, confirming the protective effect from fluoride [8,28]. This protection, however, was limited, and it could be improved if fluoride would be combined with other ingredients, such as polyvalent metal ions (like stannous) or polyphenols [3,8,28].

We also tested a solution containing the combination of fluoride with stannous ions ($Sn^{2+}$/F⁻_Rinse) as positive control. Its erosion-protective effect is related to the binding of stannous ions to the tooth surface, and the incorporation of these ions into the enamel, serving as a barrier against acids [6,28,29]. In combination with fluoride ions, the protective effect is enhanced [30,31], and our results confirm this trend, where the $Sn^{2+}$/F⁻_Rinse group presented a significantly better protective effect than the group containing only fluoride (Control_F⁻).

In the presence of the pellicle, the protective effect of the $Sn^{2+}$/F⁻_Rinse was probably not only due to the deposition of stannous and fluoride ions onto the tooth surface, but also because these ions can modify the pellicle. On the one hand, fluoride can modulate the protein composition of the pellicle, decreasing the concentration of statherin and histatin 1 [14]. This modified pellicle remains with an electrodense appearance similar to the non-modified pellicle, but it is able to better protect the tooth surface against demineralization, and even penetrate into the demineralized areas after acid attack [16]. On the other hand, stannous modifies the pellicle, possibly by interacting with albumin [32], leading to an increase in the electron density of the basal pellicle layer, which, in turn, protect the underlying tooth surface. Remarkably, this electron-dense layer can remain rather unaffected even after an erosive challenge [16].

With the solutions containing polyphenols, it is speculated that they act primarily on pellicle modification [33], because they cross-link the proteins and increase the protein-protein interaction, eventually leading to a more electron-dense and thicker pellicle. In the present study, we used polyphenol-rich plant extracts: grape seed, grapefruit seed and blueberry. Grape seed extract contains mainly purified oligomeric proanthocyanidin (OPC), including catechin and epicatechin units with partial galloylation [34]. Their mechanism of action is most probably by interacting with proline-rich proteins and statherin present in the basal layer of the pellicle [35]. Proanthocyanidin has already shown protective effects against erosion, when applied over the salivary pellicle [36]. Grapefruit seed extract contains around 98% flavonoids, specifically flavanones, and mainly naringenin and hesperidin [37,38], and blueberry extract contains anthocyanins, mainly malvidin-3-glucoside and malvidin-3-galactoside. The exact mechanism of action of grapefruit seed and blueberry are still not fully elucidated. Previous studies from our group had shown a positive effect of the extracts on pellicle modification, whereas the present results with the extracts in the No_F⁻ groups did not confirm this protective effects. We do, however, still speculate that the polyphenols in these extracts can modify the pellicle [17,36], and the presence of fluoride significantly enhanced the protective effect.

Grape_Seed_F$^-$ showed the best protective effect, better than all the controls (Control_-No_F$^-$, Control_F$^-$, and Sn$^{2+}$/F$^-$_Rinse). The mechanism of action of the solutions containing polyphenols and fluoride can be manyfold. In addition to the effect of the polyphenols on the proteins of the acquired pellicle, they can also combine with the calcium ions from saliva/pellicle and act in a similar way as the CaF$_2$-like layer, covering the enamel surface [39]. Additionally, the polyphenols could exert an inhibitory effect on proteases, inhibiting certain proteases like cathepsins, elastase and some MMPs [40–43]. Therefore, they could protect the pellicle proteins from degradation, keeping the pellicle in its original, erosion-protecting conformation. This protection could happen two-fold: directly, by inhibiting the proteases themselves and, thereby, hindering their action on protein degradation; or indirectly, by crosslinking the proteins in the pellicle and, thereby, making them less prone to the degradation by the proteases. This might especially be important for the *in vitro* model used here, as there is no fresh supply of undegraded proteins to replenish the pellicle, like there would be *in vivo*.

Interestingly, when fluoride is also present in combination with the polyphenols, we observed a considerably better protection than the solution containing only fluoride or only polyphenols. In fact, the rSH results even suggest a synergistic effect of the polyphenols with the fluoride, leading to better protective effects, especially for the grape seed extract with fluoride (Grape_Seed_F$^-$), which presented even better results than the stannous-containing solution (Sn$^{2+}$/F$^-$_Rinse). We speculate that the positive effect of this solution is related to both the polyphenols and fluoride, where the polyphenols will act on pellicle modification (crosslinking the proteins, inhibiting proteases, and binding to calcium ions in the pellicle), while the fluoride will act on pellicle modification and on the dental surface, all amassing to the protective effect observed in the results.

Remarkably, the synergistic effect between fluoride and polyphenols yielded even better protection than the fluoride-only or the positive control (stannous-containing) solutions. These results, however, should still be viewed with the caveat that this study has some limitations. Primarily, even though we used human saliva, we formed the salivary pellicle *in vitro*, which can be considerably different from those formed *in vivo* [44]. Secondly, although we used human teeth, the specimens were ground to reach a healthy (non-demineralized) layer of enamel. But teeth that already present signs of erosive tooth wear may have different susceptibilities to further demineralization [45,46]. Thirdly, the present study used an initial erosion model, where the demineralization is limited to the surface of the specimens and no bulk surface loss has occurred. Future studies should consider how the modified pellicle would perform under such severe conditions and also under abrasive impacts.

## Conclusions

We speculate that the present synergistic effect derives from multiple modes of action, where the combination of fluoride and polyphenols in a solution can have a protective effect acting both on the dental surface and on the pellicle modification. So, we conclude that polyphenols from plant extracts, when combined with fluoride, improve the protective effect of salivary pellicles against enamel erosion

## Supporting information

**S1 File. Data set of the results of the present study.**
(XLSX)

## Acknowledgments

The authors would like to thank Barbara Beyeler for her hard work in the laboratory assisting with data gathering.

## Author Contributions

**Conceptualization:** Thiago Saads Carvalho, Tommy Baumann.

**Data curation:** Thiago Saads Carvalho, Khoa Pham.

**Formal analysis:** Thiago Saads Carvalho, Khoa Pham.

**Supervision:** Thiago Saads Carvalho.

**Writing – original draft:** Thiago Saads Carvalho, Khoa Pham.

**Writing – review & editing:** Thiago Saads Carvalho, Daniela Rios, Samira Niemeyer, Tommy Baumann.

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
