## [Decision Letter · Decision Letter 0]

13 Jul 2022

PONE-D-22-03910

There is a synergistic effect between plant extracts and fluoride to protect against enamel erosion

PLOS ONE

Dear Dr. Carvalho,

Thank you for submitting your manuscript to PLOS ONE. After careful consideration, we feel that it has merit but does not fully meet PLOS ONE’s publication criteria as it currently stands. Therefore, we invite you to submit a revised version of the manuscript that addresses the points raised during the review process.

We have received comments from three reviewers. Please carefully address them all in our revised submission and response to reviewers file.

We look forward to receiving your revised manuscript.

Kind regards,

Carla Pegoraro

Division Editor

PLOS ONE

Reviewers' comments:

Reviewer's Responses to Questions

**Comments to the Author**

1. Is the manuscript technically sound, and do the data support the conclusions?

Reviewer #1: Partly

Reviewer #2: Yes

Reviewer #3: Yes

2. Has the statistical analysis been performed appropriately and rigorously? 

Reviewer #1: No

Reviewer #2: Yes

Reviewer #3: Yes

3. Have the authors made all data underlying the findings in their manuscript fully available?

Reviewer #1: Yes

Reviewer #2: Yes

Reviewer #3: Yes

4. Is the manuscript presented in an intelligible fashion and written in standard English?

Reviewer #1: Yes

Reviewer #2: Yes

Reviewer #3: Yes

5. Review Comments to the Author

Reviewer #1: This is a welcome addition to the are of erosion research. The study is sound; however, I do have quite a few comments as outlined below.

I am not a fan of ‘absolute’ manuscript titles, esp. since this was ‘just’ an in vitro study. Hence, please either rephrase it or add ‘in vitro’ to the title.

The entire manuscript would benefit from some editing. Here are just a few examples from the abstract.

Line:

23 – ‘such as’ rather than “like”; comma after “teas”

24 – ‘explored’ not “explore” (past tense throughout)

25-26 – It would be better to write ‘We distributed enamel specimens into 9 groups…”

26 – ‘F-‘ not “F-“ (entire manuscript actually; superscript “-“, this may be lost as a result of the Editorial Manager); the same applies to Sn2+ (31)

32 – ‘specimens’ not “speicmens”

32-24 – The description of the experimental procedure could be expanded. Were the cycles performed continuously without any prolonged remineralization phases after each cycle? Was it all done within one day? Even the description in the m&m section is unclear. The way it is written would suggest it was all done in one day. What is the relevance of this procedure? This would relate to five (fluoride) treatments within a roughly 8-hour period.

35 – ‘into’ not ‘to’

38 – ‘For’ not “In”

38-40 – I do not think there is a need to state that both factors affected rSH when their interaction was significant. To make it easier for the reader, perhaps the authors can add some actual rSH data? I would also suggest to leave the rSRI and calcium data to the main text and focus more on the primary outcome measure rSH, assuming it was.

45-46 – I do not think the authors can derive this conclusion from their study as the extracts were not tested in a model without salivary proteins.

48 – perhaps ‘most promising’ rather than “best”. Or perhaps leave this out entirely: “yielding the best results”.

45-48 – I would suggest something along these lines to be a more truthful conclusion: ‘We conclude that polyphenols from plant extracts, when combined with fluoride, improve the protection against enamel erosion in a model with salivary proteins.’

I would suggest moving the “experimental design” section to the front of the m&m section.

Why were the acid solutions pooled rather than analyzed individually?

How reliable are the calcium data as the treatments could have modified the pellicle layer to attract more or less calcium from the saliva? In other words, how much of the calcium that was solubilized was from the specimens vs. the pellicle?

Statistical analysis: While I would consider myself anything but an expert, I do not understand why the authors removed specimens from groups without outliers for the sole purpose to have balanced groups. This does not make sense to me. Just consider how clinical trials are being conducted. No subject would be excluded based on such a criterion. I would also like to see more detail about the procedure to test for outliers.

Results: A significant interaction between study factors does not necessarily mean what the authors stated in lines 230-232. It simply means that they interact; it could well have been that one extract enhanced the protection afforded by fluoride when another did the opposite.

Now that I have seen figure 1, I cannot see how the authors can justify their conclusion in the abstract (“We conclude that polyphenols from plant extracts seem to interact with salivary proteins, modifying the pellicle and improving the protection against enamel erosion,…”). None of the extracts were different from deionized water in the absence of fluoride.

The authors argued against the usefulness of both rSRI and calcium in the discussion and questioned their relevance. Why did the authors include these data anyway, let alone decided to include these methods during project development?

The authors’ attempt to rationalize their findings is interesting. They could have investigated calcium release from ‘extract’ treated specimens, previously exposed to saliva or not, under neutral or basic conditions (e.g., wash specimens with NaOCl and analyze the NaOCl solutions for calcium in a model without acid challenges?) to support at least one of their theories.

The last paragraph is somewhat negating a possible manuscript. It is highly unusual to do what the authors did and reminds me somewhat of a TV series… Perhaps the authors would be better off highlighting the limitations of their present study to help the reader putting this all into context?

Reviewer #2: The text is well written and the results are supported by the methodology performed. The study of plant extract as potential preventive and protective tools against erosive wear has shown interesting and promissing results by recent literature, and this study corroborates with it. However the results presented are very preliminary, and although human saliva was used, the in vitro results showed only the protective potential in initial erosion, without considering a more severe erosive cycling. Also, abrasion was not considered in the text, and shall be listened as an limitation of the work.

Reviewer #3: Title: There is a synergistic effect between plant extracts and fluoride to protect against enamel erosion

This is an interesting study regarding the synergistic effect between plant extracts and fluoride against enamel erosion. Here are some points that should be addressed:

- The surface hardness measurements only give information regarding the surface, the softening characteristics of the surface, but not the amount of tissue that was lost, which can be evaluated by measuring surface loss using for example optical profilometry. So, surface hardness can only offer qualitative information for the nature of the erosive challenge.

- It should be more informative to provide images of the surface of the specimens after the erosive challenge to evaluate also qualitatively the erosive activity on each experimental group.

- Preventive measures against erosion are usually suggested when there is early diagnosis of erosive tooth lesions and not in intact teeth. Moreover, different enamel lesion depths may have different susceptibility to demineralization. As a result, this should be mentioned as a limitation of this study.

Carvalho TS, Baumann T, Lussi A. Does erosion progress differently on teeth already presenting clinical signs of erosive tooth wear than on sound teeth? An in vitro pilot trial. BMC Oral Health 2017;17:14.

6. PLOS authors have the option to publish the peer review history of their article (what does this mean?). If published, this will include your full peer review and any attached files.

Reviewer #1: No

Reviewer #2: No

Reviewer #3: No

---

## [Author Response · Author response to Decision Letter 0]

28 Sep 2022

We provide a point-by-point response letter for the Reviewers. Please see attached document.

---

## [Decision Letter · Decision Letter 1]

31 Oct 2022

Synergistic effect between plant extracts and fluoride to protect against enamel erosion: an in vitro study 

PONE-D-22-03910R1

Dear Dr. Thiago Saads Carvalho, Privatdozent, PhD

We are pleased to inform you that your manuscript has been deemed scientifically suitable for publication and will be formally accepted for publication once it meets all pending technical requirements.

Within a week, you will receive an email detailing the necessary changes. When these have been addressed, you will receive a formal acceptance letter and your manuscript will be scheduled for publication.

An invoice for payment will follow shortly after formal acceptance. To ensure an efficient process, please log in to Editorial Manager at http://www.editorialmanager.com/pone/, click the 'Update My Information' link at the top of the page and verify that your user information is updated. till the date. If you have any questions regarding billing, please contact our author billing department directly at authorbilling@plos.org.

If your institution(s) have a press office, let them know about your upcoming article to help maximize its impact. If you are going to prepare press materials, please inform our press team as soon as possible, but no later than 48 hours after receiving formal acceptance. Your manuscript will remain under a strict press embargo until 2 p.m. ET on the date of publication. For more information, contact onepress@plos.org.

Sincerely,

Cesar Felix Cayo-Rojas, Ph.D.

Universidad Privada San Juan Bautista, Facultad de Estomatología, Lima, Perú.

Editor Académico 

PLOS ONE

---

## [Editor Report · Acceptance letter]

8 Nov 2022

PONE-D-22-03910R1 

Synergistic effect between plant extracts and fluoride to protect against enamel erosion: an in vitro study 

Dear Dr. Carvalho:

I'm pleased to inform you that your manuscript has been deemed suitable for publication in PLOS ONE. Congratulations! Your manuscript is now with our production department. 

Kind regards, 

on behalf of

Dr. César Félix Cayo-Rojas 

Academic Editor

PLOS ONE